# Morphological, Physiological and Mechanical Features of the Mutable Collagenous Tissues Associated with Autotomy and Evisceration in Dendrochirotid Holothuroids

**DOI:** 10.3390/md21030134

**Published:** 2023-02-21

**Authors:** Maria Byrne

**Affiliations:** School of Life and Environmental Sciences, University of Sydney, Sydney, NSW 2006, Australia; maria.byrne@sydney.edu.au

**Keywords:** sea cucumber, MCT, connective tissue, histology, electron microscopy, Echinodermata

## Abstract

Evisceration in dendrochirotid sea cucumbers leads to expulsion of the digestive tract, pharyrngeal complex and coelomic fluid through rupture of the anterior body wall. This process involves failure of three mutable collagenous tissue (MCT) structures, the introvert, the pharyngeal retractor muscle tendon, and the intestine-cloacal junction. These are complex structures composed of several tissue strata. The MCT in the three autotomy structures contains collagen fibrils, unstriated microfibrils, and interfibrillar molecules. Neurosecretory-like processes (juxtaligamental-type) with large dense vesicles (LDVs) are prominent in the autotomy structures. Biomechanical tests show that these structures are not inherently weak. Failure of the autotomy structures can be elicited by manipulating the ionic environment and the changes are blocked by anaesthetics. Autotomy and evisceration are under neural control, but local neural elements and neurosecretory-like processes do not appear to be a source of factors that cause MCT destabilisation. The LDVs remain intact while the tissue destabilises. The coelomic fluid contains an evisceration inducing factor indicating a neurosecretory-like mediation of autotomy. This factor elicits muscle contraction and MCT destabilisation. As the autotomy structures are completely or partially surrounded by coelomic fluid, the agent(s) of change may be located in the coelom (systemic origin) as well as originate from cells within the MCT. The biochemistry and mechanism(s) of action of the evisceration factor are not known. This factor is a promising candidate for biodiscovery investigation.

## 1. Introduction

The ability of echinoderms to rapidly alter the mechanical properties of their collagenous connective tissue through neural control is a unique feature of this marine phylum [1,2,3,4]. There are two contrasting manifestations of mutable collagenous tissue (MCT) variable tensility. These are ‘catch’, in which the specialised regions of connective tissue exhibit reversible stiffening/softening properties, and ‘autotomy’, when the tissues exhibit irreversible catastrophic failure leading to the loss of body parts [1,2,3,4,5,6,7,8,9,10,11,12,13,14,15,16,17]. In the Holothuroidea (sea cucumbers), the behaviour, physiology and morphology of the body wall catch connective tissue and its roles in posture control, protection, and defence are well studied [4,6]. The sea cucumber body wall has been used as a model system to investigate the physicochemical, molecular, and mechanistic features of echinoderm MCT [4,5]. Loss of body parts through autotomy is a striking behaviour of echinoderms and is caused by rupture of specialised MCT structures [1,2,11,12,13,14,15,16,17]. These structures are designed to break and are associated with specialised features to reduce trauma and facilitate regeneration [2,10,18,19]. The tensility changes in both catch and autotomy MCT are attributed to changes in the extracellular matrix (ECM) and interfibrillar cross links [5]. The unusual properties of MCTs and the extensive regenerative abilities to replace lost structures have undoubtedly contributed to the ecological success of echinoderms [10,20]. 

During autotomy the ECM becomes destabilised due to a loss in the cohesion between the collagen fibrils [2,3,5]. Potential agents of MCT change may be localised within the tissues (e.g., neurosecretory-like processes) as well as external to the tissue (e.g., coelomic hormone-like agents) [1,2,3,5,13,21,22,23,24,25,26,27]. Neurosecretory-like processes containing dense vesicles that vary in size and profile are characteristic of echinoderm MCT [2,5,13,16,21,26,27,28]. These cells are called juxtaligamental cells following the first use of this term for these cells in the ophiuroid arm autotomy structure [13]. These cells and agents contained in the dense vesicles are considered to be the effectors of MCT tensility change [5]. The MCT structures that rupture during ophiuroid and crinoid autotomy are well studied [2,7,11,13,21,22,26,27]. 

Evisceration in holothuroids is a dramatic form of autotomy that results in expulsion of internal organs [29,30,31,32,33,34,35,36,37,38]. It is thought that this phenomenon may be an adaptive trait to establish a physiological reset for the individual after discarding gut associated waste accumulation, to shed parasites or as a predator decoy to escape full body predation [29]. Holothuriida and Synallactida sea cucumbers (previously order Aspidochirotida), eviscerate posteriorly with expulsion of the digestive tract and associated structures and fluids (e.g., haemal vessels, respiratory trees, coelomic fluid) through the anus [33,34,35]. The tentacles and pharyngeal complex are retained. Sea cucumbers in the order Dendrochirodtida, eviscerate anteriorly with expulsion of the digestive tract along with the tentacles, pharyngeal complex, and coelomic fluid through rupture of the anterior body wall [29,30,31,32,36,37]. The cloaca and respiratory trees are retained. 

The Dendrochirotida is a diverse order of sea cucumbers distinguished by their highly branched tentacles, which they use for suspension feeding (Figure 1a) [39]. They are ecologically important, especially in areas with high levels of suspended food where aggregations of thousands of individuals can occur [29,40]. Evisceration in dendrochirotids has long attracted attention in studies that have investigated the behaviour, physiology, morphology, and mechanics of the autotomy structures [23,24,29,30,36,37]. Species in the genera *Eupentacta* and *Sclerodactyla* are notoriously prone to eviscerate (Figure 1) and have been used as model species to investigate MCT autotomy and the cell and tissue processes involved in autotomy and regeneration [14,15,16,23,24,31,32,41,42,43]. The present review provides a synthesis of the biology of dendrochirotid autotomy structures with respect to their morphological organisation, the cell and tissue processes involved with autotomy and aspects of their physiology and biomechanics. The incidence of dendrochirotid evisceration in nature and regeneration of the lost organs are reviewed elsewhere [30].

## 2. Behavioural Events

Evisceration in dendrochirotids follows a series of behavioural events that starts with whole body muscle-driven contraction and ends with expulsion of the internal organs and coelomic fluid (Figure 1). The process involves the sudden destabilisation and rupture of three autotomy structures: (1) the introvert, the dexterous anterior portion of the body wall, (2) the tendons that link the pharyngeal retractor muscles to the longitudinal body wall muscles, and (3) the intestine–cloacal junction (Figure 1a–d). The introvert is a highly flexible structure that supports the tentacles, anterior intestine, and pharyngeal complex (Figure 1a,c). The tendons form a strong junction between two major muscles to anchor the pharyngeal complex to the body wall (Figure 1b,c). The intestine ruptures along the narrow zone where it connects to the cloaca. 

Contraction of the pharyngeal retractor and longitudinal body-wall muscles generate opposing forces to facilitate separation of these muscles in parallel with tensility changes in the tendon. This is followed by whole body contraction which propels the digestive tract, associated haemal vessels, and coelomic fluid anteriorly (Figure 1d). As the hydrostatic pressure increases, the introvert extends changing from a firm opaque structure to one that is thin and translucent as it fills with the autotomised organs. The introvert finally ruptures, expelling the pharyngeal complex, water vascular ring, oral nerve ring, digestive tract, and coelomic fluid (Figure 1d). The wound is quickly sealed off by muscle contraction. Evisceration results in extensive tissue damage leading to loss of whole organ systems and a large proportion of organism biomass. 

## 3. Histology and Tissue Organisation of the Autotomy Structures

The autotomy structures, as shown here for *Eupentacta quinquesemita*, comprise complex tissue strata. They contain collagen fibrils (banding periodicity, 60–67 nm), unstriated microfibrils, ECM and a range of cellular inclusions (see below) (Figure 2a,b,e–h). Some of the unstriated microfibrils are attached to and often cross between collagen fibrils while others are isolated in the connective tissue matrix (Figure 2a,g,h).

### 3.1. Introvert

The introvert is dominated by ECM which, together with associated muscle cells, (Figure 2c and Figure 3a,b) contribute to its dexterity and viscous properties [41]. The dermis forms the bulk of the introvert and consists of an outer subepidermal layer, a thin dense connective tissue layer, and an inner prominent loose connective tissue layer (Figure 3a,b). Muscle cells are dispersed through the dermis and occupy 1–4% of the tissue in cross sectional area [42] (Figure 2c and Figure 3c). These cells likely play a role in the extension and contraction of this structure, but do not appear to play a role in autotomy. The peritoneum, body wall muscles (circular and longitudinal), subperitoneal nerve plexus, and nerve cords form the innermost tissue layer of the introvert (Figure 3a,b).

The dense connective tissue layer is a collagenous (20–160 nm diameter fibrils) stratum that spans the circumference of the introvert (Figure 3a–c). This tissue layer is likely to have reversible-stiffening (catch) properties that assist in the maintenance of tentacle posture during feeding. Cross and longitudinal sections of the collagen bands shows that the fibrils vary in diameter. Thick and thin fibrils are interspersed with each other (Figure 2a,b). The loose connective tissue layer is an electron-lucent stratum containing scattered cells suspended in what appears to be a gel-like connective tissue matrix with microfibrils fibrils (7–12 nm diameter) and collagen fibrils (40–60 nm diameter) (Figure 2f and Figure 3a,b). Bands of muscle cells supported by a collagenous sheath are also present (Figure 2c,d). Histochemical tests show that the loose connective tissue is dominated by glycosaminoglycans [42]. 

### 3.2. Pharyngeal Retractor Muscle Tendon

At their anterior end, the pharyngeal retractor muscles attach to the pharyngeal ring ossicles and at their posterior end they link with the longitudinal body wall muscles via the tendons (Figure 1b,c). The tendons form a strong link between two major muscles. They form a compact sheath of collagen fibrils (30–40 nm diameter) and microfibrils (10–15 nm diameter) surrounding and supporting the ends of the muscle cell bundles (Figure 4a,c).

### 3.3. Intestine-Cloacal Junction

The intestine-cloacal junction is typical of the holothuroid digestive tract, with the inner digestive mucosal lining, a central connective tissue layer, muscle layers, nerve plexus, and peritoneum [16,30]. The connective tissue layer is dominated by ECM that has a granular appearance reminiscent of haemal fluid. Microfibrils are abundant in this layer along with a few collagen fibrils and scattered cells (Figure 2g,h).

## 4. Cells Associated with the MCT 

### 4.1. Neuronal Profiles

Bundles of axons encased in a basal lamina are scattered through dermal layer of the introvert (Figure 5a–d). These are occasionally accompanied by an associated cell. It is not known if these are nerve cell bodies or glial-like support cells (Figure 5a,b, see also [16]). These cells are active as indicated by their cytoplasmic inclusions, including Golgi complexes, mitochondria and endoplasmic reticulum. The axons contain several types of vesicles including clear (70–80 nm diameter), dense core (80–90 nm diameter), and small dense (80–140 nm diameter) vesicles (Figure 5c,d). Some of the axon bundles in the introvert are surrounded by connective tissue and others occur in association with muscle cells. Similar bundles of axons occur in the connective tissue of the pharyngeal retractor muscle tendon, but were not observed in the intestine autotomy region. 

Neurosecretory-like cells and processes filled with large dense vesicles (LDVs), similar to the juxtaligamental-type cells typical of echinoderm MCT are prominent in the three autotomy structures (Figure 4a,b and Figure 6a–e). In some cases, these processes are surrounded by a basal lamina (e.g., Figure 6a) and so are somewhat separated from the adjacent connective tissue, but elsewhere they lack a basal lamina and are directly apposed with collagen fibrils (e.g., Figure 6d). The processes likely anastomose through the MCT as indicated by grazing sections where the middle or periphery of the processes is in view (Figure 6a). This is also indicated by the presence of processes in cross and longitudinal section in the same view (Figure 6b,d). The LDVs vary in shape from round (150–300 nm diameter) to ellipsoidal and elongate (180–550 nm × 125–220 nm) and dumbbell-shaped. These variable profiles may be due to different sections through elongated spindle-shaped or sausage-shaped granules. The variable shapes and sizes of the LDVs may also indicate different types of vesicles in different types of cells. Cells that contain dense vesicles and production of material as indicated by Golgi complex and endoplasmic reticulum activity are present in the connective tissue (Figure 6c,f). It is not known if these cells are the source of the processes that contain the LDVs. In the pharyngeal retractor muscle tendon, processes filled with LDVs, similar to those in the connective tissue occur in close association with muscle cells (Figure 4c).

### 4.2. Other Cells in the Connective Tissue

Phagocytes and morula cells are abundant in the connective tissue (Figure 3c–e and Figure 5a). Morula cells appear to function in the maintenance of connective tissue as a source of ECM [43]. Their synthetic activity involves production of matrix and fibrillar material that are released through vesicle breakdown. These cells have a mast-cell-like appearance and are particularly abundant just below the peritoneal epithelium [43]. Morulas are highly active cells, but their role in the connective tissue and the product(s) that they release are not known.

## 5. Cell and Tissue Changes That Occur during Autotomy

Ultrastructural examination of the autotomy structures of *E. quinquesemita* and the cell and tissue changes that occur during evisceration indicate a catastrophic loss in the structural integrity of the MCT [15,16]. As autotomy and evisceration are under neural control, particular attention was paid to neuronal elements in the autotomy structures. During autotomy, the ECM in the three autotomy structures takes on a hydrated-dissolved appearance. This is associated with interfibrillar slippage and disarray of the collagen fibrils and microfibrils but these fibrillar elements remain intact (Figure 7a). The cellular components normally supported by the connective tissue (e.g., axons, neurosecretory-like processes, muscle cells, morula cells) become physically damaged and torn and disperse into the coelom (Figure 7d,e,h). 

The abundance of ECM in the introvert is important for autotomy because it facilitates distension during evisceration. The dermis changes from a gel-like to a fluid-like state. When the introvert expands to accommodate the coelomic fluid and viscera it undergoes viscid flow due to loss of tensility. The collagen fibrils slide past each other as the introvert distends. Destabilisation of the introvert appears to start from the internal (coelomic-side) surface potentially associated with infiltration of agents in the coelom (see below) and continues into the tissue as a wave of disruption. During expansion of the introvert the axons, neurosecretory-like processes, and muscle cells extend along the direction of distension and detach from the tissue (Figure 7c,d). The cell membranes rupture, but the LDVs appear to remain intact, although some neurons have a swollen appearance (Figure 7b). 

During autotomy, the pharyngeal muscle tendon changes from a compact collagenous structure to one in disarray as the ECM breaks down. The peritoneum peels away and the muscle cells disintegrate (Figure 7f). Many neurosecretory-like processes remain intact through autotomy. The vesicle-filled neurons associated with muscle cells maintained a strong attachment despite tissue disruption (Figure 7f). During tendon autotomy, axons from the dissociated nerve plexus and neurosecretory-like processes appear swollen and disrupted, but the contents of the LDVs in these cells do not appear to have changed (Figure 7g). 

Overall, neurosecretory-like processes morphologically similar to the juxtaligamental cells that are characteristic of echinoderm MCTs are prominent in the autotomised tissues. The LDVs contained in these processes appear largely unchanged while the tissue breaks down. There is little to no morphological evidence that the LDVs are a source of agents that cause MCT change. 

## 6. Mechanical Properties of the Autotomy Structures

Constant stress creep tests were used to quantify the mechanical properties of isolated introvert preparations of *E. quinquesemita* [41]. The creep rate (slope of the creep curve) divided by the stress used provided a measure of viscosity. As expected, these tests showed that the introvert is a highly extendable viscous structure. This structure exhibited marked deformation (up to 900% increase in length) under stress, likely due to the dominance of ECM in the dermis. This result also indicated that the connective tissue fibrils slide past each other as the tissue deforms, along with a reduction in fibre-matrix adhesions. The introvert did not exhibit autotomy-like softening and failure in the tests and so does not have an inherent weakness to explain its behaviour during evisceration.

Isolated pharyngeal muscle tendon preparations extended under a constant load either failed in the middle of the muscle or where the tendon inserts into the pharyngeal ossicle. Thus, the tendons of *E. quinquesemita* are mechanically strong, as also shown in similar breaking tests of the tendons of *Sclerodactyla briareus* [24,41]. 

Autotomy of the *E. quinquesemita* MCTs can be mimicked in vitro by manipulating the ionic composition of bathing solutions [41]. The absence of Ca^++^ and Mg^++^ in test solutions makes the introvert highly compliant decreasing viscosity. In contrast, increased Ca^++^ stiffens the introvert, increasing viscosity. The presence of increased K^+^ results in rapid failure of isolated introvert preparations at low strain values, as is also the case for the retractor muscle tendon. The tendon is also destabilised by Mg^+ +^ free seawater. 

## 7. Induction and Control of Evisceration and the Endogenous Evisceration Factor

Evisceration can be elicited by various noxious stimuli such as increased temperature, low salinity, hypoxia, and pinching with forceps, stimuli that may mimic deleterious conditions in the environment or a predator attack [23,24,36,37]. This response can also be elicited by increasing the K^+^ concentration of the coelom. In isolated tendon preparations excess K^+^, elicits strong contraction of the retractor and body wall muscles creating tension on the tendon which then softens and ruptures [40]. Potassium-induced autotomy of the tendon MCT also occurs in the absence of muscle contraction, as shown in anaesthetised muscle-tendon preparations [41]. Strong muscle contraction induced by acetylcholine is not accompanied by autotomy [41]. Thus, muscle contraction alone does not cause tendon autotomy. It is suggested that K^+^ may exert an indirect effect by stimulating cells in the MCT that control tensility or has a direct effect disrupting the stability of the ECM [41]. Evisceration of whole animals and rupture of isolated autotomy structures is elicited by electrical stimulation and this response is blocked by anaesthesia. These physiological investigations and anaesthetic block of autotomy indicate that MCT destabilisation is under neural control [24,41]. 

An evisceration factor that induces autotomy is present in the perivisceral coelomic fluid expelled by *E. quinquesemita* and *S. briareus* [23,24]. Injection of the fluid expelled during evisceration elicits evisceration in intact individuals. In contrast, normal fluid harvested from the perivisceral coelom of intact individuals does not. The evisceration factor has been isolated from several tissues with the highest levels in haemal and peritoneal (coelomic epithelial lining) tissue extracts. Low concentrations of the evisceration factor cause a slow rhythmic contraction of the retractor muscle [24]. In higher concentrations this factor induces rapid tendon autotomy in isolated preparations [23,24]. Thus, the evisceration factor elicits two responses-muscle contraction and MCT destabilisation, both of which are blocked by anaesthesia. These observations indicate that the evisceration factor may be a complex of neurochemical-like molecules with multi-functional roles. This factor is presumably a cellular product and may originate in epithelial cells, nerve plexus cells or cells in the coelomic fluid. Nerve-like processes, containing dense vesicles, located in the peritoneal lining are suggested to be a potential source of the evisceration factor [24]. The evisceration factor can also be isolated from the peritoneum of species that do not eviscerate indicating that it has functions other than its involvement in evisceration [23]. The reduction in introvert thickness is caused by increased coelomic pressure, and the evisceration factor may work in concert with this pressure to cause failure of the introvert. Factors that affect the mechanical properties of the body wall MCT of *Stichopus chloronotus* have also been isolated from the coelomic fluid of this species [44]. 

The evisceration factor is a heat stable molecule and gel fractionation of peritoneal extracts from *E. quinquesemita* indicate that the active fraction has a molecular mass approximating 30–50 amino acid residues [23]. This active fraction induces autotomy in isolated tendon preparations. Similar isolation of the evisceration factor of *S. briareus* indicated that it is a single small molecule (150 D) [24]. It appears that the evisceration factor is a small peptide and likely to be a neuropeptide. Without knowledge of the chemical composition of this factor, we can only speculate as to its mode of action. 

As the autotomy structures of sea cucumbers are completely or partially surrounded by perivisceral coelomic fluid, the agents of change may originate from two sources, cells within autotomy structures and/or from a systemic origin as indicated by the presence of an evisceration factor in the coelom. The evisceration factor may be secreted from activated cells to effect autotomy. Active factors in the perivisceral coelom may also be released from damaged tissues and act as agents to enhance the response. A potential sequence of events with involvement of a neurosecretory-like cascade in the cell and tissue responses is shown in Figure 8. In this scenario the evisceration stimulus may induce activity in upstream neurons or act directly on evisceration-factor-producing cells resulting in release of this factor into the coelom. Once in the perivisceral coelom, the evisceration factor may act as a direct effector of connective tissue change or may act as a messenger activating effector cells. This is a neurosecretory, hormone-like mechanism whereby the effector substance is secreted into a transport medium to effect change in distant target cells and tissues. Considering that the peritoneum is a major source of evisceration factor in *E. quinquesemita* and *S. briareus*, disruption of this layer and its associated nerve plexus may result in the release of factors that modulate autotomy. 

## 8. Discussion

As characteristic of echinoderm autotomy [1,2,3,5,11], the structures that rupture during evisceration contain MCT that undergoes rapid destabilisation, leading to failure. As shown for the autotomy structures of other echinoderms, the introvert and retractor muscle tendon of dendrochirotids do not have a pre-existing mechanical weakness to account for their failure during autotomy [2,23,41]. Dendrochirotid autotomy structures are histologically complex, being composed of several tissue types (nervous, muscle, connective tissue) and so the processes underlying MCT autotomy and eventual evisceration are also complex. Morphologically, the collagen and microfibrillar network in the autotomy structures of dendrochirotids are typical of echinoderm connective tissues, as also are the neurosecretory-like (juxtaligamental) processes [1,13,26,27,28]. During evisceration, the fibrillar elements in the autotomy tissues remain intact, but they and the cells associated with them disperse in an apparently ‘dissolving-hydrating’ ECM. Not all dendrochirotids exhibit evisceration [30]. It would be interesting to compare the morphology of the introvert and retractor muscle tendon of species that do and do not eviscerate. In addition to the MCT response, evisceration involves a whole-body muscle contraction, which accelerates the failure of the autotomy structures and ejection of the viscera. 

For ophiuroids there is strong evidence that the juxtaligamental cells form a link between the nervous system and the MCT, although the mechanism underlying this link remains elusive [5]. The LDVs in the juxtaligamental cells are considered to be the local effectors of mutability [1,4,5,21,26,27,45]. Several studies suggest that active MCT-change factors reside in the dense vesicles [1,5,13,21,27]. Changes in the density and integrity of the vesicles during autotomy, the release of the granules into the extracellular matrix, exocytotic profiles that indicate release of granule content, and a reduction in their number are suggested to indicate the release of autotomy promoting factors [7,9,21,27,45]. However, these changes may also be a consequence of mechanical disruption during MCT destabilisation. Changes in the density of LDVs are also suggested to be associated with the variable states of tensility in catch MCT [5]. 

For *E. quinquesemita*, there is little or no evidence of change in the LDVs or any neuronal vesicular inclusion during failure of the autotomy structures. Remarkably, despite being displaced as the tissues around them fall apart, including detachment into the coelom, the neurosecretory-like processes and their LDVs remain largely intact with no apparent change in vesicle contents. This suggests that these vesicles are not a local source of agents that cause connective tissue change. There may be other cells that control MCT tensility within the autotomy structures, but potential candidates were not identified. Similarly, some categories of neurosecretory-like LDVs remain intact through asteroid arm autotomy [9].

The LDVs in the juxtaligamental-type processes in the autotomy MCT of *E. quinquesemita* exhibited considerable morphological variability. They are likely to encompass several vesicle types in diverse cells with tissue-specific (e.g., muscle, connective, nervous) functions. Some of the juxtaligamental-type processes are surrounded by a basal lamina as also seen for similar processes in the body wall of *E. quinquesemita* [46], but other processes did not have an associated basal lamina. It is not clear if the dense vesicle-containing processes in the autotomy MCTs of *E. quinquesemita* can be separated into the two categories designated for echinoderm MCT (type 1, spherical vesicles 100–200 nm diameter; type 2, ellipsoidal vesicles 200–1000 nm length/diameter) [5]. The profiles of the dense vesicle in the juxtaligamental-type cells of *E. quinquesemita* indicate that they are the result of sections through elongate or tortuously shaped (e.g., dumbbell profiles) vesicles. Moreover, these variably shaped vesicles were also present in the same cell. The juxtaligamental-type cells in dendrochirotid autotomy connective tissue may represent a diversity of cell types and may differ from those seen in other echinoderms. Several juxtaligamental-type cells have also been observed in the sea cucumber body wall dermis and in the Cuvierian tubules of *Holothuria forskali* [47,48]. These studies also localized tensilin to these cells. Tensilin is a molecule that stiffens the MCT and is involved in the connective tissue catch response [49,50,51].

Processes containing LDVs such as those seen in the connective tissue were also associated with retractor muscle cells. These are similar to those associated with the longitudinal body wall muscles of *Apostichopus japonicus* [52]. They are likely to be neuromuscular specialisations [52]. Coordinated neuromuscular activity is an essential component of evisceration, the control of which is probably different from that causing MCT tensility change and may involve the muscle-associated LDV-containing processes. It is difficult to discern which LDV-containing processes function only with respect to MCT in *E. quinquesemita*. The product in the LDVs may also have a dual role in controlling muscle contractility as well as MCT tensility. 

In contrast to holothuroid autotomy, the autotomy structures of ophiuroids and crinoids are dominated by extracellular tissue and lack muscle cells making interpretation of the role of the juxtaligamental cells more straight forward [7,21]. In ophiuroids and crinoids, these cells are the only morphologically detectable source of agents that may be linked to connective tissue change [7,13,21,26,27,45]. Moreover, the autotomy structures of ophiuroids and crinoids are located at a distance from coelomic vessels and so the potential for hormonal translocation is minimal. Like dendrochirotid autotomy structures, asteroid arm autotomy structures also consist of several tissue types including muscle cells and are influenced by a coelomic factor [8,9,25].

As found for other studies of echinoderm MCTs [1,3,48,49,50,53,54], variable tensility of isolated autotomy structures of *E. quinquesemita* can be mimicked in vitro by manipulating the ionic composition of test solutions. Change in the ionic environment may influence the MCT tensility by altering the interactions between the ECM and fibrillar elements [41] or may trigger an ion-dependent cellular responses [5]. As found here, excess K^+^ also reduces the viscosity of asteroid autotomy structures [9]. Studies testing the response of isolated crinoid autotomy structures using neurochemical agents provide evidence that neural transmission is involved in MCT destabilisation [22].

The perivisceral coelom plays a multifunctional role in evisceration, as a source of agents that cause MCT tensility change and muscle contraction and in providing hydrostatic pressure for introvert expansion. The presence of a coelomic evisceration factor suggests involvement of neurosecretory or hormone-like activity with cells involved in autotomy located at a distance from the MCT structures and use coelom as a conduit for an agent(s) that effects change. The heat resistance of the evisceration factor suggests a peptidergic neurosecretory/hormonal agent [23,24]. The autotomy-promoting factor of asteroids also occurs in the coelomic fluid and has a peptide component [25]. Peptides are also implicated in tensility change in holothuroid body wall catch MCT [55].

With the histological complexity of their autotomy structures, their elaborate behavioural response, and their large perivisceral coelom, it is not surprising that autotomy in the Holothuroidea differs from this phenomenon in other echinoderms. There are, however, fundamental parallels in the MCT autotomy response. Like other echinoderms, dendrochirotid autotomy is controlled by an unconventional relationship between the nervous-neurosecretory system and extracellular tissue. The mechanisms at the neuronal level are not known. There is strong evidence for involvement of active factors distributed through the coelom that have a direct transmitter-like or neurosecretory-like mode of operation. It seems that important insights into echinoderm autotomy and the tensility change in the MCT would be generated using the sea cucumber model. We know very little about MCT autotomy that occurs during evisceration in holothuriid and synallactid sea cucumbers. This is a major gap in knowledge.

The mutable properties of echinoderm connective tissues are remarkable. Despite considerable advances in our understanding of the biology of catch and autotomy MCTs, there remain key knowledge gaps as to how the unusual tensility changes are accomplished and the physiological mechanisms involved [5]. In particular, the molecular biology of the interfibrillar matrix and the cohesion between the connective tissue fibrils and the ECM are key to understand [5]. 

The notorious ability of holothuroids to discard internal organs has attracted attention for 100+ years [38], but the rationale and processes underlying this phenomenon and why some species do, and others do not eviscerate, are not understood. The presence of an endogenous evisceration factor(s) in dendrochirotids has been known for some time [23,24], but its chemical nature and mode of action are not known. Filling this gap in our knowledge is essential to understanding holothuroid autotomy. We do not know whether the active component is one or several molecules. The evisceration factor is likely to include unidentified neurotransmitters or neuropeptides. The distinct activity of the evisceration factor in eliciting MCT tensility change as well as muscle contraction is intriguing and points to its potential as a biodiscovery molecule(s) for biomedical applications, as highlighted for other aspects of echinoderm MCT [5]. This factor is likely to be endogenous across echinoderms. Characterisation of the molecular nature and mode of action of the evisceration factor is a high priority for research. The high activity of the peritoneal cell layer points to a potential source to investigate. The morula cells which are abundant along the peritoneum are also a good candidate (see [56]). With emerging single cell-seq technology, characterising the expression profiles and secretory nature of the peritoneal cells and morula cell seems feasible. 

## Figures and Tables

**Figure 1 marinedrugs-21-00134-f001:**
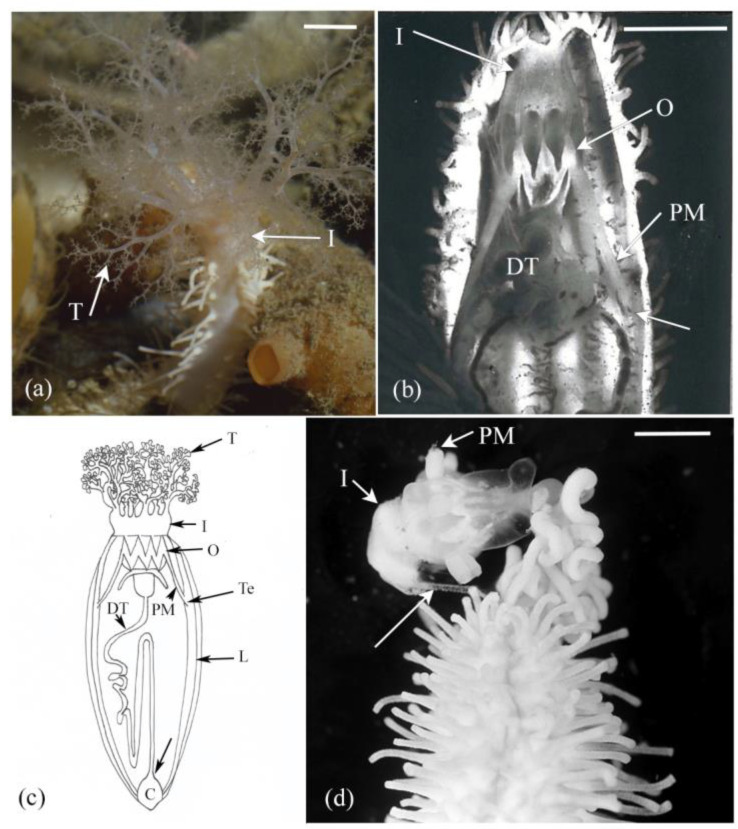
(**a**) In situ image of *Eupentacta quinquesemita* showing the tentacles (T) that are used in suspension feeding and the supporting introvert (I). (**b**) Dissected individual showing the ossicles (O) of the pharyngeal complex, the introvert (I), the digestive tract (DT), and the pharyngeal retractor muscle (PM) and where it attaches (arrow) to the longitudinal body wall muscle. (**c**) Diagram showing the three autotomy structures; the introvert (I), pharyngeal retractor muscle (PM), tendon (Te), and the intestine-cloacal junction (arrow). C, cloaca; DT, digestive tract; L, longitudinal body wall muscle; O, ossicle ring; T, tentacles. (**d**) Evisceration involves softening of the introvert (I), which eventually dissolves into strands (arrow), note the broken end of the pharyngeal retractor muscles (PM). Scales: (**a**) = 1 cm (**b**,**c**) = 0.5 cm.

**Figure 2 marinedrugs-21-00134-f002:**
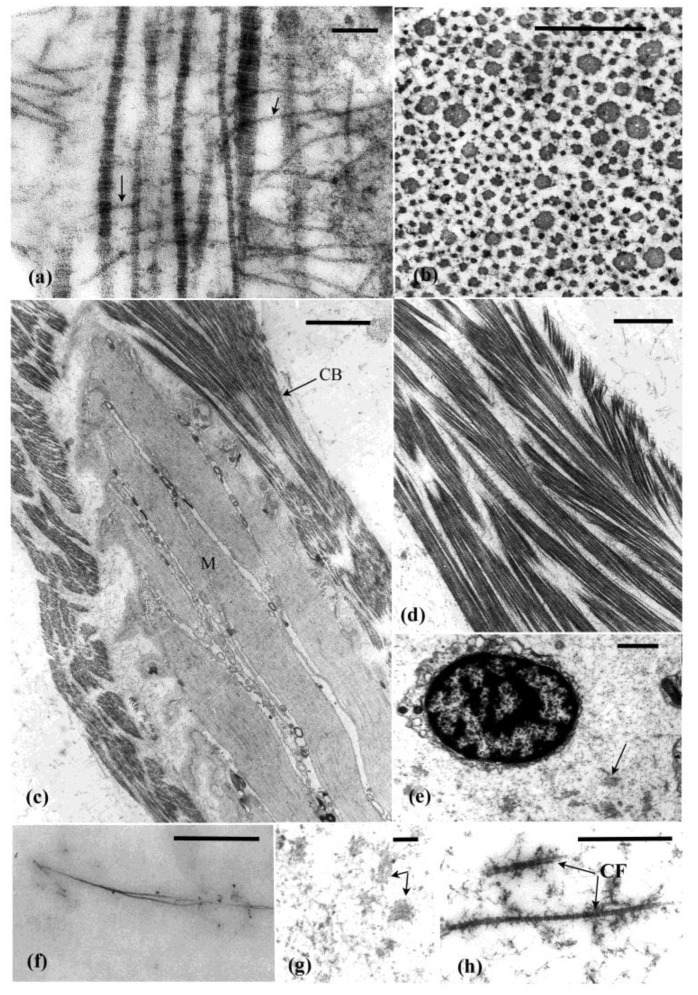
Collagenous connective tissues of *Eupentacta quinquesemita* autotomy structures (**a**). Longitudinal section of collagen fibrils that differ in thickness and the unstriated fibrils (arrows) that cross between them. (**b**) Cross section showing collagen fibrils that differ in thickness. (**c**) Muscle bundle in the introvert connective tissue is surrounded by a band (CB) of collagen fibrils. (**d**) Collagen band that surrounds the muscle bundle. (**e**) Gut connective tissue layer with cells and small aggregations of material (arrow). (**f**) Unstriated fibril in introvert connective tissue. (**g**) Gut connective tissue layer with material (arrows) that may be coagulated haemal fluid. (**h**) Collagen fibrils (CF) in gut connective tissue layer. Scales: (**a**) = 0.1 µm, (**b**) and (**d**) = 0.5 µm, (**c**) = 2 µm, (**e**) = 1 µm (**f**–**h**) = 0.5 µm.

**Figure 3 marinedrugs-21-00134-f003:**
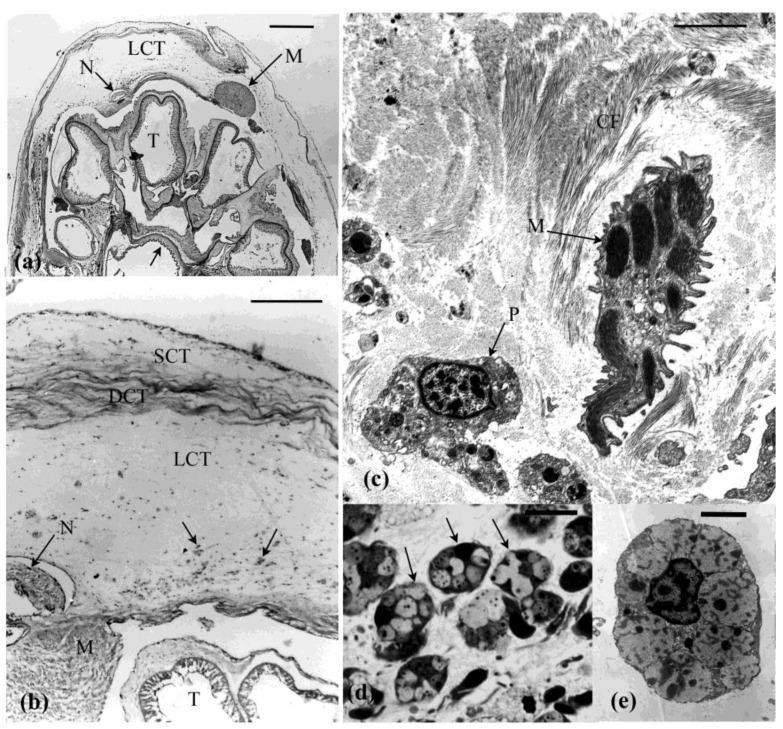
Light (**a**,**b**) and transmission electron micrographs (**c**) of the intact introvert. (**a**,**b**) In cross section the introvert comprises a subepithelial connective tissue (SCT), a thin dense connective tissue layer (DCT), and is dominated by the inner loose connective tissue layer (LCT). The lumen of the tentacles (T), the radial nerve (N), and longitudinal body wall muscle (M) are evident. Arrow, oesophagus. (**c**) The dense connective region consists of collagen fibrils (CF), muscle (M) and occasional phagocytes (P). (**d**) Light micrograph showing morula cells (arrows) abundant in the subperitoneal connective tissue. (**e**) Electron micrograph of a morula cell filled with secretory vesicles. Scales: (**a**) = 2 mm, (**b**) = 0.2 mm (**c**–**e**) = 3 µm, (**d**) = 10 µm.

**Figure 4 marinedrugs-21-00134-f004:**
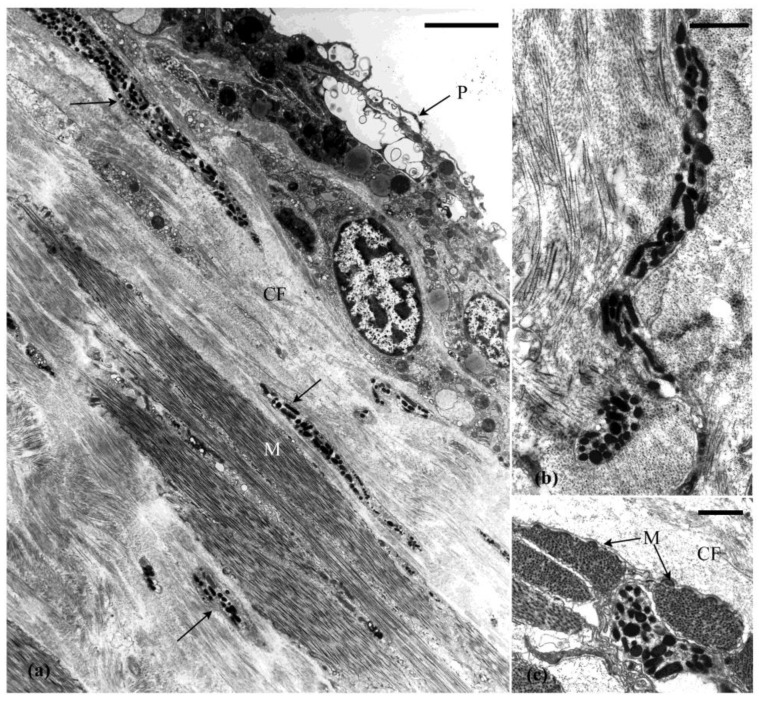
(**a**–**c**) Transmission electron micrographs of the pharyngeal retractor muscle at the tendon connection to the body wall muscle. Collagenous tissue (CF) surrounds bundles of muscle cells (M. Processes containing large dense vesicles (arrows) in the connective tissue with some associated with muscle cells. P, peritoneum. Scales: (**a**) = 3 µm, (**b**,**c**) = 1 µm.

**Figure 5 marinedrugs-21-00134-f005:**
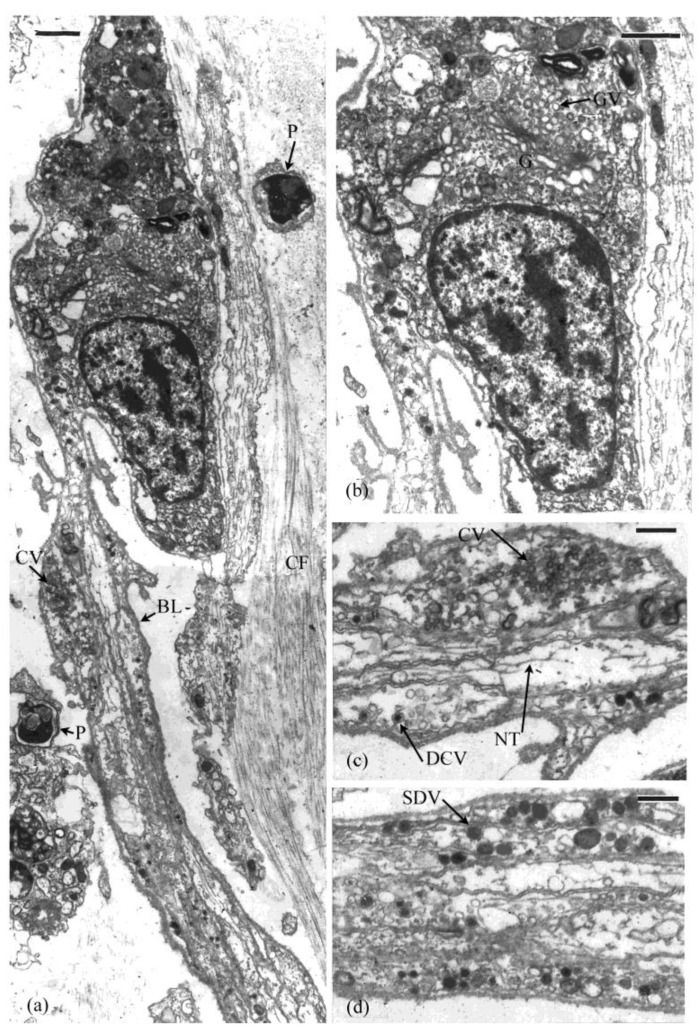
(**a**–**d**) Transmission electron micrographs of a nerve bundle in the introvert connective tissue. The nerve bundle and associated cell are encased in a basal lamina (BL) and the axons contain clear vesicles (CV), small dense vesicles (SDV), dense core vesicles (DCV), and neurotubules (NT). CF, collagen fibrils; G, Golgi complex; GV, Golgi vesicles; P, phagocyte. Scales: (**a**,**b**) = 1 µm, (**c**,**d**) = 0.5 µm.

**Figure 6 marinedrugs-21-00134-f006:**
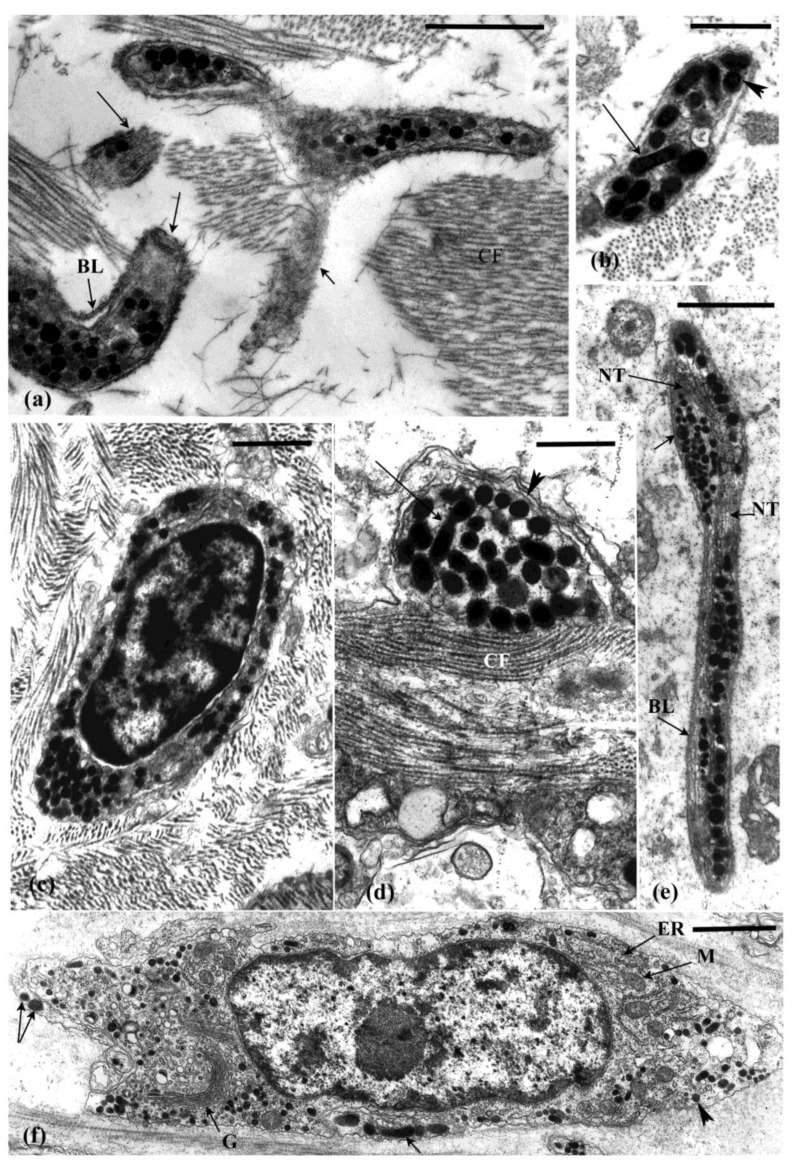
Transmission electron micrographs of processes and cells containing dense vesicles that vary in size and shape. (**a**) Process in the dense connective tissue layer of the introvert where tangential sections (arrows) indicate that these profiles may be a part of an anastomosing process. Note the basal lamina (BL) cover. (**b**) Elongated (arrow) and round (arrowhead) vesicles in the same process in the introvert. (**c**) Cell body containing dense vesicles the introvert. (**d**) Dense vesicles apposed to collagen fibrils (CF) in the tendon. (**e**) Dense vesicle process in gut connective tissue. (**f**) Cell body in the tendon connective tissue containing dense vesicles that range in size and profile (arrows, round to elongate vesicles). G, Golgi complex, ER endoplasmic reticulum, M, mitochondrion. (**a**,**c**,**f**) = 1 µm, (**b**,**d**,**e**) = 0.5 µm.

**Figure 7 marinedrugs-21-00134-f007:**
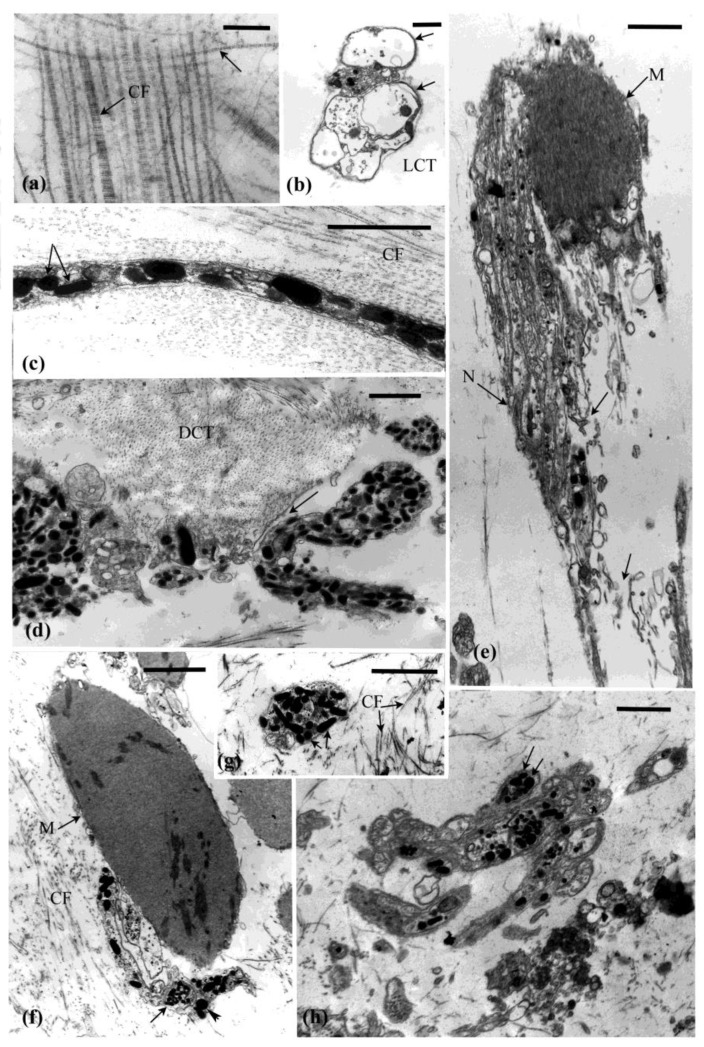
Transmission electron micrographs of autotomising MCT. (**a**) Collagen and unstriated fibrils in the introvert indicate some disarray but are intact. (**b**) Nerve-like profile in the loose connective tissue (LCT) with swollen cells (arrows). (**c**) Large dense granule processes in the dense connective tissue. Arrows show round and elongated vesicles. (**d**) Large dense granule processes in the dense connective tissue (DCT) tearing away (arrow) from the surrounding collagen, but the granules are intact. (**e**) Nerve bundle (N) and muscle cell (M) in introvert with tears evident (arrows). (**f**) Degenerating pharyngeal muscle cell (M) and associated axon-like processes remaining attached despite tissue disruption. Arrowhead shows round and elongated vesicles adjacent to each other. (**g**) Dense vesicles processes in the tendon connective tissue (**h**) Autotomised gut autotomy site with large and small granules (arrows) intact among disrupted tissue (**a**) = 0.3 µm (**b**) = 0.5 µm (**c**–**e**) = 1 µm, (**f**–**h**) = 2 µm.

**Figure 8 marinedrugs-21-00134-f008:**
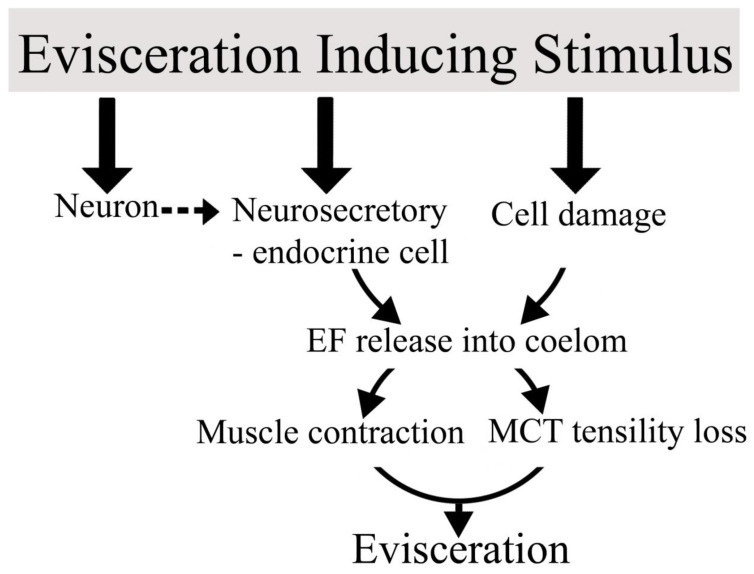
Hypothetical model of the sequence of events leading to autotomy and evisceration with involvement of a neurosecretory-like cascade and cell and tissue responses. It is proposed that the inducing stimulus (e.g., sea star predator) activates neurons which stimulate evisceration factor (EF) producing neurosecretory-endocrine cells resulting in release of EF into the coelomic fluid or directly activates EF producing cells. The process may also result in cell damage releasing EF into the coelom.

## Data Availability

Not applicable.

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
