# Peer review of "Morphological, Physiological and Mechanical Features of the Mutable Collagenous Tissues Associated with Autotomy and Evisceration in Dendrochirotid Holothuroids"

_marinedrugs, 2023, doi:10.3390/md21030134_

Round 1

Reviewer 1 Report

I have no special comments on the content of the manuscript. I would advise the author to combine the chapters "Results" and "Discussion" into one chapter "Results and discussion". And to make a small general conclusion.

 Minor remarks:

Line 55

retained, –>  retained.

 Line 60

[2-3,5] –> [2,3,5]

Line 64

[2,5.13,16,21,37] –> [2,5,13,16,21,37]

It is also better to cite more recent works on JLC of crinoids here:

DOI 10.1086/BBLv226n2p81; DOI 10.1007/s00435-020-00483-4;

Line 122

Collagen fibrils in gut connective tissue layer –> Collagen fibrils (CF) in gut connective tissue layer.

 Line 183

diam –> diameter

 Line 334

In what units was weight measured? Da or kDa?

 Line 404

Stichopus –> Apostochopus japonicus

Author Response

REVIEWER 1 COMMENTS:

Editorial comments

  1. Line 55 retained, –>  retained.
  2. Line 60 [2-3,5] –> [2,3,5]
  3. Line 64 [2,5.13,16,21,37] –> [2,5,13,16,21,37]
  4. Line 122 Collagen fibrils in gut connective tissue layer –> Collagen fibrils (CF) in gut connective tissue layer.
  5. Line 183 diam –> diameter
  6. Line 404 Stichopus –> Apostochopus japonicus

All these changes have been made.

Other comments

  1. Line 334 In what units was weight measured? Daor kDa? Thanks for spotting this error - changed "(150 D)”
  1. It is also better to cite more recent works on JLC of crinoids here:  DOI 10.1086/BBLv226n2p81; DOI 10.1007/s00435-020-00483-4; 

These papers are now cited and incorporated in the text.

Reviewer 2 Report

The review by Maria Byrne is an important summary of the current state of knowledge on visceral autotomy in dendrochirotid holothurians. Sea cucumber species of the order Dendrochirota are of extreme interest from the standpoint of fundamental regenerative biology, as they can quickly grow back most of their organ systems following injury. As one of the founding fathers of the echinoderm MCT biology elegantly put it once, autotomy is often a prelude to regeneration in echinoderms. In dendrochirotids, autotomy is as dramatic as  their regenerative capacities. These animals are capable of discarding the entire oral body end along with most of the visceral organs.

The review summarizes the structural organization and cellular composition of the pre-determined autotomy zones that undergo irreversible loss of tensile strength in response to a trigger. The author then discusses possible cellular and molecular mechanisms involved in the control and execution of the autotomy.

I  have only a few suggestions on how this excellent work can be improved:

* The author makes a point that events in the mutable collagenous tissue (MCT) that lead to evisceration in holothurians might be different from those happening during autotomy in other echinoderm classes. In particular, the vesicle content of neurosecretory (juxtaligamental) cells does not  seem to be released during the loss of tensile strength by the MCT structures in dendrochirotids. In brittle stars, the juxtaligemental cells are considered to be the primary source of the effector molecules that, upon exocytosis, directly influence the strength of interfibrillar cohesion in collagen bundles. Recent studies (Demeuldre et al., 2017; Bonneel et al., 2022) localized one of the known effector molecules -- tensilin -- to the sea cucumber neurosecretory cells and showed that it controls the MCT stiffness. Also, a survey of sequence databases (Mittal et al., 2022) suggests that tensilin is present in all five extant echinoderm classes. These recent findings suggest a degree of conservation in the mechanisms controlling the MCT tensile strength among echinoderms.

* In addition to photographs in Fig. 1, it would be nice to have a schematic diagram showing the basic sea cucumber anatomy and the position of the three autotomy MCT structures that the author describes. This would help a reader who might not have an extensive background in sea cucumber biology.

* The Fig. 5 caption is missing definitions for "G" and "GV", which probably stand for the Golgi body and Golgi vesicles, respectively

* What do the arrow and the arrowhead indicate in Fig. 7f?

Author Response

REVIEWER 2 COMMENTS:

Editorial comments

I thank the reviewer for the supportive comments in the first two sections of this review.

Specific comments
I  have only a few suggestions on how this excellent work can be improved:

  1. The author makes a point that events in the mutable collagenous tissue (MCT) that lead to evisceration in holothurians might be different from those happening during autotomy in other echinoderm classes. In particular, the vesicle content of neurosecretory (juxtaligamental) cells does not  seem to be released during the loss of tensile strength by the MCT structures in dendrochirotids. In brittle stars, the juxtaligemental cells are considered to be the primary source of the effector molecules that, upon exocytosis, directly influence the strength of interfibrillar cohesion in collagen bundles. Recent studies (Demeuldre et al., 2017; Bonneel et al., 2022) localized one of the known effector molecules -- tensilin -- to the sea cucumber neurosecretory cells and showed that it controls the MCT stiffness. Also, a survey of sequence databases (Mittal et al., 2022) suggests that tensilin is present in all five extant echinoderm classes. These recent findings suggest a degree of conservation in the mechanisms controlling the MCT tensile strength among echinoderms. 

This is a good point and important findings.  I have mentioned this in the discussion and cited the Demeuldre et al., 2017; Bonneel et al., 2022) and the location of tensillin to the juxtaligamental cells.  As this manuscript is about autotomy and tensilin functions in catch so not quite on the topic I have kept this new text brief. The text has been changed to:

“The juxtaligamental-type cells in dendrochirotid autotomy connective tissue may represent a diversity of cell types and may differ from those seen in other echinoderms. Several juxtaligamental-type cells have also observed in the sea cucumber body wall dermis and in the Cuvierian tubules of Holothuria forskali [47,48]. These studies also localized tensilin to these cells. Tensilin is a molecule that stiffens the MCT and is involved in the connective tissue catch response [49-51].”

  1. In addition to photographs in Fig. 1, it would be nice to have a schematic diagram showing the basic sea cucumber anatomy and the position of the three autotomy MCT structures that the author describes. This would help a reader who might not have an extensive background in sea cucumber biology. Good suggestion - there is a new figure - Fig 1c, a diagram that shows the anatomy and position of the autonomy tissues
  1. The Fig. 5 caption is missing definitions for "G" and "GV", which probably stand for the Golgi body and Golgi vesicles, respectively

Correction made

  1. What do the arrow and the arrowhead indicate in Fig. 7f?

Good spotting, the information has been added:

“Arrowhead show round and elongated vesicles adjacent to each other.”

Reviewer 3 Report

1. The figures provide sufficient quality (except Fig 8 which should be revised) but the sources are not determined.

2. In addition to the available figures, graphical figures can be implemented to enhance the representation of the mechanisms.

3. Abbreviations section is missing from the manuscript.

4. The study should add additional information about Dendrochirotid Holothuroids (e.g., taxonomy) in the manuscript.

5. The abstract of the manuscript requires revision in terms of fluency and writing or grammatical errors.

6. The punctuation should be improved throughout the whole text.

7. In lines 85 and 86 the word "compliant" is unclear. Please revise.

8. The sentence in line 137 should be revised.

9. It is better to abbreviate "large dense vesicles" as "LDVs" not "LDV".

10. Dumbbell-shaped is correct in line 198.

11. "Elongated" in line 274 seems to be correct. 

12. The sentence in lines 330 and 331 should be revised.

13. "Molecular weight ~150" in line 334 should be revised.

14. The sentence in lines 354 and 355 should be revised.

15. "Effect" in lines 385, 406, and 428 should be revised.

16. The sentence in lines 442, 443, and 444 should be revised.

17. The sentence in line 449 should be revised.

18. "Rational" in line 454 should be revised

Author Response

REVIEWER 3 COMMENTS:

1.The figures provide sufficient quality (except Fig 8 which should be revised) but the sources are not determined.

Figure 8 has been redone so that the quality is now sufficient.  As this figure was drafted for this manscript there is not a previous source.

  1. In addition to the available figures, graphical figures can be implemented to enhance the representation of the mechanisms.

Following the suggestion of reviewer 2 I have added an anatomical schematic of showing the location of the autotomy structures

  1. Abbreviations section is missing from the manuscript.

Two abbreviations have been deleted (LCT, DCT) with the words now spelled out. The remaining abbreviations used in the main body of the text - ECM. LDV and MCT are routine for the topic of this journal issue and they are defined first use.  There are also not many others so I feed a glossary is not needed.

  1. The study should add additional information about Dendrochirotid Holothuroids (e.g., taxonomy) in the manuscript.

This is a good point. I have added two references one being a taxonomic review of the Holothuroidea. The new sentences are:

“The Dendrochirotida is a diverse order of sea cucumbers distinguished by the highly branched tentacles that they use for suspension feeding (Fig. 1a) [39]. They are ecologically important, especially in areas with high levels of suspended food where aggregations of thousands of individuals can occur [29,40].” 

New references:

Rowe, F.W.E.; O’Hara. T.D.; Bardsley, T.M.; Class Holothuroidea. In: Byrne, M,; O’Hara. T.D.; Australian Echinoderms: Biology Ecology and Evolution. 2017, CSIRO Press 447-490.

Costello, J.; Keegan, B.F.; Feeding and related morphological structures in the dendrochirote Aslia lefevrei(Holothuroidea: Echinodermata). Mar. Biol. 1984, 84, 135-142.

Minot editorial comments - All of these corrections have been made

  1. The abstract of the manuscript requires revision in terms of fluency and writing or grammatical errors.
  2. The punctuation should be improved throughout the whole text.
  3. In lines 85 and 86 the word "compliant" is unclear. Please revise.

Changed to “flexible”

  1. It is better to abbreviate "large dense vesicles" as "LDVs" not "LDV".
  2. Dumbbell-shaped is correct in line 198.
  3. "Elongated" in line 274 seems to be correct. 
  4. "Effect" in lines 385, 406, and 428 should be revised.
  5. Rational" in line 454 should be revised
  6. ."Molecular weight ~150" in line 334 should be revised.

Changed to (150 D)”

Sentence revisions

The sentence in line 137 should be revised. This is now 2 sentences:

“Cross and longitudinal sections of the collagen bands shows that the fibrils vary in diameter. Thick and thin fibrils are interspersed with each other (Fig. 2a,b)."

The sentence in lines 330 and 331 should be revised This is now 2 sentences:

“The evisceration factor is a heat stable molecule and gel fractionation of peritoneal extracts from E. quinquesemita indicate that the active fraction has a molecular mass approximating 30–50 amino acid residues [23]. This active fraction induces autotomy in isolated tendon preparations.”

The sentence in lines 354 and 355 should be revised. This sentence has been shortened:

“As characteristic of echinoderm autotomy [1-3,5,11], the structures that rupture during evisceration contain MCT that undergoes rapid destabilisation, leading to failure.”

The sentence in lines 442, 443, and 444 should be revised. This is now 2 sentences:

“There is strong evidence for involvement of active factors distributed through the coelom that have a direct transmitter-like or neurosecretory-like mode of operation. It seems that important insights into echinoderm autotomy and the tensility change in the MCT would be generated using the sea cucumber model.”

 The sentence in line 449 should be revised. The paragraph has been revised:

“The mutable properties of echinoderm connective tissues are remarkable. Despite considerable advances in our understanding of the biology of catch and autotomy MCTs, there remains key knowledge gaps as to how the unusual tensility changes are accomplished and the physiological mechanisms involved [5]. In particular, the molecular biology of the interfibrillar matrix and the cohesion between the connective tissue fibrils and the ECM are key to understand [5].”

Round 2

Reviewer 3 Report

All of my comments and corrections are considered.

Author Response

thank you very much for your Comments